# CONSTRAINT-AWARE ZERO-SHOT VISION-LANGUAGE NAVIGATION IN CONTINUOUS ENVIRONMENTS

## ABSTRACT

We present Constraint-Aware Navigator (CA-Nav), a zero-shot approach for Vision-Language Navigation in Continuous Environments (VLN-CE). CA-Nav reframes the zero-shot VLN-CE task as a sequential constraint-aware sub-instruction completion process, continuously translating sub-instructions into navigation plans via a cross-modal value map. Central to our approach are two modules namely Constraint-aware Sub-instruction Manager (CSM) and Constraint-aware Value Mapper (CVM). CSM defines the completion criteria of decomposed sub-instructions as constraints and tracks navigation progress by switching sub-instructions in a constraint-aware manner. Based on the constraints identified by CSM, CVM builds a value map on-the-fly and refines it using superpixel clustering to enhance navigation stability. CA-Nav achieves the state-of-the-art performance on two VLN-CE benchmarks, surpassing the compared best method by 12% on R2R-CE and 13% on RxR-CE in terms of Success Rate on the validation unseen split. Furthermore, CA-Nav demonstrates its effectiveness in real-world robot deployments across diverse indoor scenes and instructions[1].

## 1 INTRODUCTION

Vision-Language Navigation (VLN) is a fundamental task in Embodied AI. It requires the agent to navigate in novel environments according to natural language instructions (Anderson et al., 2018b). Early efforts focused on discrete environments, where the agent follows instructions to navigate on predefined connectivity graphs. Recently, the more practical VLN in Continuous Environments (VLN-CE) (Krantz et al., 2020) has garnered increasing attention, allowing the agent to navigate freely in 3D environments. However, most existing VLN and VLN-CE methods (Chen et al., 2022b; An et al., 2024) rely on annotated trajectories for policy learning, encountering issues of data scarcity and generalization. In response, zero-shot VLN has emerged as a promising direction, leveraging Vision-Language Models (VLMs) (Li et al., 2023) and Large Language Models (LLMs) (Achiam et al., 2023) for decision-making, without training on annotated trajectories. Existing methods (Zhou et al., 2024b; Chen et al., 2024) follow a text-based prompt paradigm, converting visual observations and navigation history into text, which is then combined with the full instruction and input into an LLM to infer the next action.

Despite these attempts, few works have explored zero-shot VLN-CE. One approach would be to directly adapt existing zero-shot VLN methods to continuous environments. However, such an adaptation faces two major challenges and shows substantial performance degradation. First, continuous environments expand the state space, making it difficult for the agent to accurately track navigation progress and determine which part of the instruction is being executed. Existing methods (Zhou et al., 2024b; Chen et al., 2024) that input the full instruction into the LLM might overlook the importance of monitoring the completion status of sub-instructions. Second, converting visual observations into text often leads to losing visual details and environmental structures (Wunderlich & Gramann, 2021), impairing the agent's spatial understanding and path planning.

In light of the above, we present Constraint-aware Navigator (CA-Nav), a new approach for the challenging zero-shot VLN-CE task. CA-Nav reframes VLN-CE as a sequential constraint-aware sub-instruction completion process. Within each episode, a Constraint-aware Sub-instruction Manager (CSM) decomposes instructions and switches between sub-instructions by assessing whether

---

[1]Project webpage with demonstration videos and code https://anony-mouser.github.io/CA-Nav/

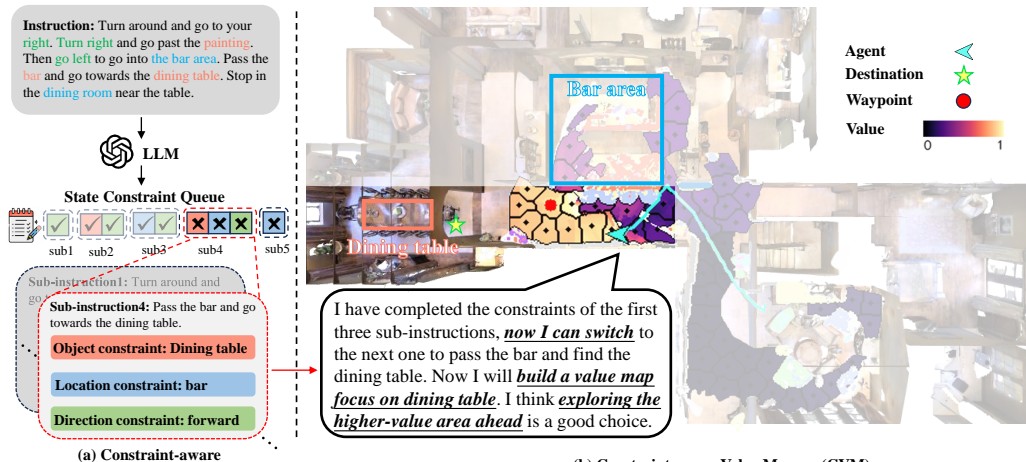

Figure 1: Illustration of the proposed CA-Nav. (a) The Constraint-aware Sub-instruction Manager decomposes the instruction into a sequence of sub-instructions and identifies object constraints, location constraints and direction constraints for each of them. (b) During navigation, a Constraint-aware Value Mapper builds a value map based on the landmark prompt provided by CSM and uses the superpixel clustering method to segment it into regions. It switches sub-instructions in a constraint-aware manner and chooses the most promising region's geometric center as waypoints.

the relevant constraints are met. Meanwhile, a Constraint-aware Value Mapper (CVM) builds and continuously updates a value map based on current constraints and observations, capturing both visual details and spatial layouts. CA-Nav generates navigation plans using CVM, which are then executed by classical control algorithms, guiding the agent to accomplish each sub-instruction until the episode terminates.

As shown in Figure 1 (a), at the start of an episode, CSM prompts the LLM to decompose the instruction into sub-instructions and generate constraints for completing each sub-instruction. During navigation, CSM continuously monitors the fulfillment of these constraints and switches to the next sub-instruction once the current constraints are met. We leverage VLMs (Liu et al., 2023; Li et al., 2023) to detect various constraints, such as landmark detection, location recognition, and direction estimation. These constraints are composed to cover diverse sub-instruction expressions.

Given the sub-instructions identified by CSM, the next challenge is grounding them in the map alongside visual details for improved spatial understanding and navigation planning. We address this with the CVM, which evaluates the potential of each observation for satisfying the current constraint. As illustrated in Figure 1 (b), CVM builds a map to capture both the semantics and spatial layout of the environment. Using a VLM, it calculates the similarity between current observations and landmarks associated with task constraints, generating a constraint-aware value map projected onto the ground plane. To enhance the accuracy and stability of navigation, superpixel clustering (Achanta et al., 2012) is applied to refine the map, reducing noise and maintaining coherence within regions. This enables the agent to select waypoints from high-value regions, ensuring that navigation aligns with both task constraints and environmental understanding. Classical control algorithms (Sethian, 1999) is finally used for the waypoint navigation.

Experiments in both simulation and real-world demonstrate the effectiveness of CA-Nav. CA-Nav achieves the state-of-the-art on two VLN-CE benchmarks in the zero-shot setting, surpassing the compared best methods by 12% on R2R-CE and 13% on RxR-CE in terms of success rate on the validation unseen split. Notably, CA-Nav achieves approximately 10 times faster response times and a 95% cost reduction compared to counterparts. Furthermore, real-world robot deployments verify CA-Nav's potential for practical applications, showcasing its effectiveness across open-vocabulary instructions and various indoor environments.

## 2 RELATED WORK

**Vision-Language Navigation.** Vision-Language Navigation (VLN) has garnered significant attention in recent years. Various methods have been explored: some employ novel architectures and cross-modal alignment techniques (An et al., 2021; Chen et al., 2021b; Wang et al., 2023b), some

utilize data augmentation (He et al., 2021; 2024; Chen et al., 2022c; Li et al., 2022; Lin et al., 2023; Wang et al., 2023d; Li & Bansal, 2024), and some explore pre-training methods and auxiliary tasks (Hao et al., 2020; Qiao et al., 2022; 2023; Hong et al., 2024). However, these methods are limited to discrete environments and show a significant performance drop when applied to the real-world (Anderson et al., 2021). Consequently, Krantz et al. (2020) transfers the VLN task into continuous environments (VLN-CE) with low-level actions. This more practical task setting promotes the development of sim-to-real for VLN (Zhang et al., 2024; Wang et al., 2024b).

Early approaches to VLN-CE concentrate on supervised learning. Some of them directly learn low-level control (Krantz et al., 2021; Raychaudhuri et al., 2021; Irshad et al., 2022; Chen et al., 2022a; Wang et al., 2023c; He et al., 2023), while others use a waypoint predictor (Hong et al., 2022) trained on the navigation graph from Matterport 3D dataset (Chang et al., 2017) to discretize the environment into candidate waypoints. This allows models trained for VLN can be transferred to VLN-CE (Hong et al., 2021; An et al., 2023; 2024). However, these methods rely heavily on annotated trajectories, which demand substantial human effort to create. Therefore, we aim to develop a zero-shot approach for the VLN-CE task relying on foundation models.

**Foundation Models for Robotic Navigation.** Foundation models are those trained on broad data that can be adapted to a wide range of downstream tasks including LLMs (Brown et al., 2020; Achiam et al., 2023; Touvron et al., 2023) and VLMs (Radford et al., 2021; Li et al., 2023; Liu et al., 2023). Their strengths in reasoning, task planning, visual grounding, and multi-modal understanding make them promising for robotic navigation.

Recently, zero-shot VLN methods (Long et al., 2023; Zhou et al., 2024b;a; Chen et al., 2024; Zhan et al., 2024; Lin et al., 2024) have emerged, they use VLMs to describe observations and GPT-4 to make step-by-step decisions in discrete environments. Specifically, DiscussNav (Long et al., 2023) employs multiple GPT-4 experts to discuss the current observations, status, and instructions before moving; NavGPT (Zhou et al., 2024b) utilizes GPT-4 to process descriptions of visual observations and navigation history before navigating; MapGPT (Chen et al., 2024) converts a topological map into prompts and then uses GPT-4 for navigation. However, few studies focus on zero-shot VLN-CE. A$^2$Nav (Chen et al., 2023) attempts to decompose instructions into action-specific object navigation sub-tasks. However, it's not truly training-free as it relies on room region bounding box annotations from HM3D (Ramakrishnan et al., 2021) to collect data for training five action-specific navigators. In contrast, we introduce a new training-free method, CA-Nav, which enables entirely zero-shot sub-instruction switching in a constraint-aware manner.

## 3 METHOD

**Problem Definition.** We address the zero-shot VLN-CE task (Krantz et al., 2020), where the agent navigates to a destination following natural language instructions. Unlike methods like A$^2$Nav (Chen et al., 2023), which are trained on datasets for specific navigation skills, our approach only utilizes foundation models for decision-making. The agent is equipped with an odometry and an egocentric RGB-D camera with a 79° Horizontal Field of View (HFOV). It can perform low-level actions such as MOVE FORWARD (0.25m), TURN LEFT/RIGHT (30°), and STOP. An episode is considered successful if the agent stops within a certain distance from the target.

### 3.1 METHOD OVERVIEW

As illustrated in Figure 2, for each episode, CSM and CVM work coherently to execute instruction-following navigation. The CSM identifies key constraints that define the completion criteria for each sub-instruction, ensuring constraint-aware sub-instruction switching and navigation progress tracking. Upon completing a sub-instruction, CSM uses these constraints to automatically transit to the next one (§ 3.2). Then the CVM uses the identified constraints to build a constraint-aware value map, which guides navigation for the current sub-instruction (§ 3.3).

### 3.2 CONSTRAINT-AWARE SUB-INSTRUCTION MANAGER

The Constraint-aware Sub-instruction Manager (CSM) aims to decompose instructions and track the navigation progress through explicit sub-instruction switching. We achieve this by appropriately prompting an LLM and designing a constraint-aware switching mechanism.

Figure 2: An overall pipeline of CA-Nav. The details of the Constraint-aware Value Map Generation are shown in Figure 3.

**Instruction Decomposition.** As shown in Figure 1, at the beginning of an episode, CSM decomposes the instruction into a sequence of sub-instructions. Each of them outlines the goal for the current sub-instruction and specifies the constraints for switching to the next one. We categorize these constraints as object constraints (e.g., "chair"), location constraints (e.g., "bedroom"), and direction constraints (e.g., "turn left"). Particularly, object and location constraints describe the landmarks that can be observed along the desired navigation path. Thus, they will prompt subsequent building of the value map (§ 3.3). In practice, we implement the above decomposition and constraints extraction process through an LLM, and the prompts are detailed in Appendix (§ A.1).

**Sub-instruction Switching.** In the zero-shot VLN-CE setting, the agent should be aware of its navigation progress and automatically switch sub-instructions. We achieve this through a constraint-aware sub-instruction switching mechanism. As shown in Figure 2, CSM maintains a queue in the order of decomposed sub-instructions, with each element containing a set of constraints. These constraints can be a combination of object, location, and direction constraints, depending on the current sub-instruction. CSM always selects the first unsatisfied constraint set as the current and only switches to the next set once all constraints in the current set are satisfied. The agent sequentially checks each constraint within the current set at each step (pseudocode is in § A.2). We leverage VLM and odometry information to design satisfaction checks for the constraints:

- Object Constraints: Since the navigation instructions include open vocabulary, we use Grounding DINO (Liu et al., 2023) to detect objects. A constraint is considered satisfied if the agent detects the object within a certain range $r$ which could empirically be set as 5 meters.
- Location Constraints: Indoor location detection can be approached as scene recognition, using a Visual Question Answering (VQA) model such as BLIP2 (Li et al., 2023) with the template: "Can you see the $< location >$?". If the answer is yes, the constraint is considered satisfied.
- Direction Constraints: The change in direction should be assessed based on the agent's trajectory rather than just its orientation. To do this, we query the odometry for the poses within a certain time window $\tau$, which could be empirically selected as 5. The poses are denoted as $\mathbf{p_t}$ and $\mathbf{p_{t-\tau}}$. The change of direction and angle is then calculated using their cross-product and dot product.

Sometimes, the agent may get stuck on a single constraint or switch between constraints too frequently. To address this, we empirically establish a maximum step threshold of 25 to prompt the agent to switch constraints when progress stalls, along with a minimum step threshold of 10 to ensure adequate focus on each constraint before switching.

## 3.3 CONSTRAINT-AWARE VALUE MAPPER

After CSM provides the current sub-instruction along with its associated constraints, these constraints serve as guiding factors for the value map construction. Specifically, we propose a Constraint-aware Value Mapper (CVM) to ground the constraints within the visual environment, en-

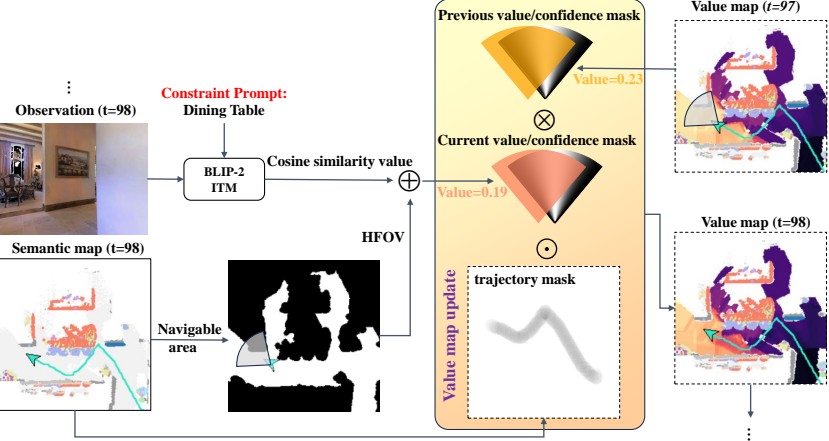

Figure 3: Details of the Constraint-aware Value Map Generation.

suring that the value map reflects the potential of each observation to satisfy the current constraints. The CVM is then refined using a superpixel clustering method for waypoint selection.

**Constraint-aware Value Map Generation.** The constraint-aware value map generation process first creates a semantic map (Chaplot et al., 2020) that captures the environment's layout and then projects navigation values onto the navigable areas. As shown in Figure 2, we first build a semantic map and extract navigable areas using egocentric RGB-D images and the camera's pose. Then we use BLIP-2 (Li et al., 2023), a pre-trained VLM to compute the cosine similarity value between the current RGB observation and the constraint prompt identified by CSM (§ 3.2). The value measures how relevant the current HFOV is for satisfying the constraint prompt. These values are then updated onto the navigable area, forming the value map.

As shown in Figure 3, when updating the value map, we focus on the current HFOV and assign the value from BLIP-2 to the value mask. The confidence mask then applies a cosine-weighted average to adjust the update. Specifically, in the confidence mask the pixels along the optical axis have a full confidence of 1, while those at the left and right edges have a confidence of 0. We set the confidence of a pixel at position (i, j) as: $c_{i,j} = \cos^2(\theta/(\theta_{\text{hfov}}/2) \cdot \pi/2)$, where $\theta$ is the angle between the pixel and the optical axis, $\theta_{\text{hfov}}$ is a constant angle HFOV. The process is formulated as follows:

$$v_{i,j}^{\text{curr}} = (c_{i,j}^{\text{curr}} v_{i,j}^{\text{curr}} + c_{i,j}^{\text{prev}} v_{i,j}^{\text{prev}})/(c_{i,j}^{\text{curr}} + c_{i,j}^{\text{prev}}); \quad c_{i,j}^{\text{curr}} = \left[ (c_{i,j}^{\text{curr}})^2 + (c_{i,j}^{\text{prev}})^2 \right]/(c_{i,j}^{\text{curr}} + c_{i,j}^{\text{prev}}) \quad (1)$$

where $v_{i,j}^{\text{curr}}$ and $c_{i,j}^{\text{curr}}$ represent current step's value and confidence at position (i, j), respectively and $v_{i,j}^{\text{prev}}$ and $c_{i,j}^{\text{prev}}$ denote previous step's value and confidence.

It is worth noting that when CSM switches constraints, the constraint prompt changes. If the previous value map is cleared entirely, the agent loses navigational cues and must rebuild from scratch, often causing it to linger and collide. To address this, we introduce a historical decay factor $\gamma$ that retains past value map $\mathbf{V}_t$ with exponentially reduced weights, helping the agent focus on new constraints while still leveraging past exploration:

$$\mathbf{V}_{t+1} = \begin{cases} \gamma \cdot f(\mathbf{V}_t), & \text{if switch constraint} \\ f(\mathbf{V}_t), & \text{otherwise} \end{cases} \quad (2)$$

where $f$ denotes the update function for the value map (pseudocode is in Appendix § A.2).

To encourage exploration, we introduce a trajectory mask representing the agent's willingness to explore. This mask starts as a matrix of ones, but its values decay exponentially by a factor of $\lambda$ in regions the agent has already traversed. The value map is then adjusted by element-wise multiplication with this trajectory mask.

**Superpixel-based Waypoint Selection.** Next, we need to choose a waypoint according to the value map. A common approach is using frontier-based exploration (FBE) (Yamauchi, 1997) which has been widely used in object navigation task (Yokoyama et al., 2024; Zhou et al., 2023; Shah et al., 2023). Among them, VLFM (Yokoyama et al., 2024) is the most comparable to ours, as it also constructs a value map and selects the frontier with the highest value as the navigation target. However, it only focuses on the boundaries of the explored area, restricting the full utilization of the value

map. We propose that using the full value map can benefit navigation, however, our CSM introduces sub-instruction switching, which may lead to abrupt value changes at the frontiers. To that end, we propose a superpixel-based waypoint selection approach that considers the global value map.

Specifically, we employ SLIC (Achanta et al., 2012) to refine the constraint-aware value map. Given current value map $\mathbf{V}$, SLIC produces a set of superpixels $\{\mathbf{S_1}, \mathbf{S_2}, \cdots, \mathbf{S_n}\}$, where each superpixel $\mathbf{S_i}$ represents a visually consistent region. Let $v(p)$ denote the value at pixel $p$ in $\mathbf{V}$, then compute the average value of each superpixel $\mathbf{S_i}$. The optimal region $\mathbf{S}^*$ is then selected based on the highest average value, guiding the agent towards areas of greater semantic relevance:

$$\mathbf{V}(\mathbf{S_i}) = \frac{1}{|\mathbf{S_i}|} \sum_{p \in \mathbf{S_i}} v(p), \quad \mathbf{S}^* = \arg\max_{\mathbf{S_i}} \mathbf{V}(\mathbf{S_i}) \tag{3}$$

Finally, the waypoint is the geometric center of the optimal region $\mathbf{S}^*$. Note that when the agent reaches the final sub-instruction, it extracts the target's segmentation mask using RepViT-SAM (Wang et al., 2023a; 2024a), which achieves real-time segmentation of anything. The mask is then projected onto the semantic map and its geometric center will be the destination waypoint (details in § A.2). After determining the waypoint, the Fast Marching Method (FMM) (Sethian, 1999) is used to plan low-level actions to the waypoint.

## 4 EXPERIMENTS AND RESULTS

### 4.1 EXPERIMENT SETUP

**Dataset and Evaluation.** We conduct experiments using Habitat simulator (Savva et al., 2019) on the val-unseen split of R2R-CE (Krantz et al., 2020) and RxR-CE (Ku et al., 2020), the only two VLN-CE datasets. They provide 1839 and 3669 step-by-step trajectory-instruction pairs across 11 val unseen environments, respectively, with RxR containing more detailed instructions.

We use standard metrics following (Anderson et al., 2018b; Krantz et al., 2020): Navigation Error (NE), i.e., the mean distance from the final location to the destination, Success Rate (SR), i.e., the proportion of episodes with NE under 3 meters, Oracle Success Rate (OSR), i.e., SR with an oracle stop policy, Success weighted by Trajectory Length (SPL), i.e., SR normalized by trajectory length, Normalized Dynamic Time Warping (NDTW), i.e., the fidelity between the agent's and the annotated trajectories, and Success-weighted Dynamic Time Warping (SDTW), i.e., NDTW weighted by SR.

### 4.2 MAIN RESULTS

**Methods for Comparison.** We compare CA-Nav with VLN-CE methods that also use low-level actions. These methods can be categorized into two types: the first type is training-based methods, including Sara (Irshad et al., 2022), Seq2Seq (Krantz et al., 2020), VLN↻BERT (Hong et al., 2022), AG-CMTP (Chen et al., 2021a), WS-MGMap (Chen et al., 2022a) and LAW (Raychaudhuri et al., 2021). The second type is zero-shot methods, including NavGPT-CE (Zhou et al., 2024b), A$^2$Nav (Chen et al., 2023) and our CA-Nav. NavGPT was originally for zero-shot VLN, we transfer it to zero-shot VLN-CE by using a waypoint model (Hong et al., 2022) to predict navigable nodes in panoramas, discretizing the continuous environment for high-level navigation planning. We also list other methods that rely on high-level actions, including Sim2Sim (Krantz & Lee, 2022), GridMM (Wang et al., 2023c), ETPNav (An et al., 2024), and BEVBert (An et al., 2023).

**R2R-CE Dataset.** As shown in Table 1, our method surpasses various models trained with panorama in NE, SR and OSR. This indicates that building an egocentric zero-shot VLN-CE system with LLMs and VLMs is feasible and potential. We believe our method's success with egocentric observations lies in the value map's dual role: storing environmental layout memory and leveraging VLM's prior knowledge. To study the performance of zero-shot methods in continuous environments, we transfer NavGPT[2] to continuous environments (i.e., NavGPT-CE), with results showing a success rate drop of over 50%. One potential issue is that the caption model sometimes misses crucial details like landmarks and spatial layout. Additionally, the LLM may generate hallucinations when summarizing navigation history or reasoning about the agent's status, resulting in incorrect navigation planning. (visualization analysis in § 4.4). The method closest to our setting is A$^2$Nav, which also uses LLM efficiently and utilizes egocentric observation. Our method surpasses it because A$^2$Nav focuses more on actions. However, descriptions of actions in instructions are more

---

[2]NavGPT achieved a navigation success rate (SR) of 34.0 and a success weighted by path length (SPL) of 42.0 on the R2R dataset.

Table 1: Comparison with SOTA methods on R2R val-unseen split. In the Efficient LLM Usage column: **-** means the LLM is not used, ✓ means the LLM is only used before navigation starts, and ✗ means the LLM is accessed for each navigation decision. †: Our reproduced NavGPT-CE for VLN-CE. ∗: Methods use the same waypoint predictor proposed in (Hong et al., 2022). ✗: $A^2$Nav pretrained the navigator on an action-specific dataset built from HM3D (Ramakrishnan et al., 2021)

| Method | Zero-shot | Efficient LLM Usage | Egocentric Obs | NE↓ | SR↑ | OSR↑ | SPL↑ |
|---|---|---|---|---|---|---|---|
| Sasra | ✗ | - | ✗ | 8.32 | 24.0 | - | 22.0 |
| Seq2Seq | ✗ | - | ✗ | 7.77 | 25.0 | 37.0 | 22.0 |
| AG-CMTP | ✗ | - | ✗ | 7.90 | 23.1 | 39.2 | 19.1 |
| VLN↺BERT | ✗ | - | ✗ | 7.66 | 23.2 | - | 21.7 |
| VLN↺BERT* | ✗ | - | ✗ | 5.74 | 44.0 | 53.0 | 39.0 |
| Sim2Sim* | ✗ | - | ✗ | 6.07 | 43.0 | 52.0 | 36.0 |
| GridMM* | ✗ | - | ✗ | 5.11 | 49.0 | 61.0 | 41.0 |
| ETPNav* | ✗ | - | ✗ | 4.71 | 57.0 | 65.0 | 49.0 |
| BEVBert* | ✗ | - | ✗ | 4.57 | 59.0 | 67.0 | 50.0 |
| WS-MGMap | ✗ | - | ✓ | 6.28 | 38.9 | 47.6 | 34.3 |
| NavGPT-CE† | ✓ | ✗ | ✗ | 8.37 | 16.3 | 26.9 | 10.2 |
| $A^2$Nav | ✗ | ✓ | ✓ | - | 22.6 | - | 11.1 |
| **CA-Nav** | ✓ | ✓ | ✓ | **7.58** | **25.3** | **48.0** | **10.8** |

Table 2: Comparison with SOTA methods on RxR-Habitat val-unseen split (only English).

| Method | Zero-shot | Efficient LLM Usage | Egocentric Obs | NE↓ | SR↑ | SPL↑ | NDTW↑ | SDTW↑ |
|---|---|---|---|---|---|---|---|---|
| LAW | ✗ | - | ✗ | 11.04 | 10.0 | 9.0 | 37.0 | 8.0 |
| VLN↺BERT* | ✗ | - | ✗ | 8.98 | 27.1 | 23.7 | 46.7 | - |
| GridMM* | ✗ | - | ✗ | 8.42 | 36.3 | 30.1 | 48.2 | 33.7 |
| ETPNav* | ✗ | - | ✗ | 5.64 | 54.8 | 44.9 | 61.9 | 45.3 |
| WS-MGMap | ✗ | - | ✓ | 9.83 | 15.0 | 12.1 | - | - |
| $A^2$Nav | ✗ | ✓ | ✓ | - | 16.8 | 6.3 | - | - |
| **CA-Nav** | ✓ | ✓ | ✓ | **10.37** | **19.0** | **6.0** | **13.5** | **5.0** |

prone to ambiguity. For example, a path to the living room might be described as either "Turn left to the living room." or "Turn slightly right, then turn left immediately and go to the living room.". We conclude that rigid execution of actions such as turning tends to cause serious cumulative errors, and the agent should focus on more clearly described instructions such as landmarks. Overall, CA-Nav achieves state-of-the-art performance in SR, NE, and OSR, and performs comparably on SPL.

**RxR-CE Dataset.** Table 2 presents the results on the RxR dataset. Instructions in RxR are much longer and contain more fine-grained descriptions of landmarks and actions, leading to more frequent instruction switches and more challenging constraints identification. Under this circumstance, CA-Nav still exceeds several models trained with panoramic observations. Moreover, our method outperforms $A^2$Nav in terms of SR and NE, and is on par with it in SPL. This indicates that CA-Nav also adapts well to complex, lengthy instructions.

## 4.3 ABLATION STUDY

We conduct ablation experiments on the R2R dataset to evaluate each component of CA-Nav, focusing on four aspects: (1) the impact of constraints, (2) value map update methods, (3) the effectiveness of superpixel-based waypoint selection, and (4) generalization to different LLMs.

**The effect of different constraints.** We begin our investigation by examining how the performance of CA-Nav is influenced by different types of constraints. The first three rows of Table 3 show the results after ablating each type of constraint individually. We observe that each constraint is crucial for CA-Nav, especially the object constraint. This further supports our analysis in § 4.2 that the agent relies more on landmarks than directions during navigation, which also explains why CA-Nav outperforms $A^2$Nav. Then, we remove all constraints except those related to the final sub-instruction, making the task more akin to object navigation. The results are presented in the fourth row of Table 3. The success rate drops by about 21% compared to our best performance, as shown in the last row. This suggests that by designing appropriate constraints and using a Constraint-aware Sub-instruction Manager, our method can autonomously switch between sub-instructions in long-

Table 3: The effect of different constraints.

| Method | NE↓ | SR↑ | OSR↑ | SPL↑ |
|---|---|---|---|---|
| w/o direction constraint | 7.74 | 24.0 | 46.9 | **10.9** |
| w/o object constraint | 8.10 | 20.9 | 45.4 | 7.2 |
| w/o location constraint | 7.93 | 23.1 | 46.6 | 8.6 |
| w/o all constraints | 7.95 | 20.0 | 36.4 | 9.8 |
| **w/ all constraints** | **7.58** | **25.3** | **48.0** | 10.8 |

Table 4: Influence of value map update methods.

| Method | NE↓ | SR↑ | OSR↑ | SPL↑ |
|---|---|---|---|---|
| None | 7.68 | 22.3 | 38.9 | 10.3 |
| w/ trajectory mask | 7.65 | 24.6 | 41.3 | **10.9** |
| w/ historical decay | **7.57** | 24.6 | 45.6 | 10.8 |
| **trajectory + historical** | 7.58 | **25.3** | **48.0** | 10.8 |

Table 5: Different waypoint selection methods.

| Method | NE↓ | SR↑ | OSR↑ | SPL↑ |
|---|---|---|---|---|
| FBE-based | 8.08 | 21.9 | **50.2** | 10.4 |
| Pixel-based | 7.87 | 22.9 | 42.9 | 10.4 |
| **Superpixel-based** | **7.58** | **25.3** | 48.0 | **10.8** |

Figure 4: CA-Nav vs. NavGPT-CE.

Table 6: Generalization to different LLMs.

| Method | NE↓ | SR↑ | OSR↑ | SPL↑ |
|---|---|---|---|---|
| GPT-3.5 | 7.66 | 21.1 | 45.0 | 9.4 |
| Claude-3.5 Sonnet | **7.41** | 25.2 | 47.1 | **11.8** |
| **GPT-4** | 7.58 | **25.3** | **48.0** | 10.8 |

horizon navigation tasks. This not only compensates for VLFM's (Yokoyama et al., 2024) inability to handle sequential tasks but also avoids frequent calls to LLMs like NavGPT.

**How can the value map be better updated?** Recall that in the value map generation section § 3.3, we designed a historical decay mechanism to leverage past explorations and a trajectory mask to encourage exploration. We investigate their influence in Table 4. In Row1, we ablate both historical decay and trajectory mask which means the value map will be completely reset upon constraint switching. It's not unexpected that this brings a severe performance drop. In Row 2, we ablate the trajectory mask, the performance drops and we observe that the agent tends to get stuck in narrow spaces such as corridors. This suggests that the trajectory mask contributes positively to the agent's exploration. In Row 3, where the historical decay is ablated and shows a worse performance. This highlights the inefficiency of discarding all previous knowledge after each constraint switches, as it forces the agent to rediscover previously explored areas. However, Row 4, which incorporates both a trajectory mask and long-term value maintenance using historical decay, effectively balances exploration with exploitation and achieves the best performance.

**Comparison of different waypoint selection methods.** To verify the effectiveness of the superpixel-based waypoint selection method, we compare it with the FBE-based method mentioned in § 3.3. We also compare it to the Pixel-based method, which directly selects the pixel with the highest value as the navigation target. Results in Table 5 show that the superpixel-based method outperforms the FBE-based method by approximately 15.5% in terms of NE, SR, and SPL. This is because value map updates tend to be uneven, making the FBE-based method, which focuses on local frontiers more susceptible to disruptions from sudden value changes (§ 4.4). While a confidence mask is applied to perform cosine-weighted averaging during value map generation (§ 3.3), the trajectory mask leads to the map's lack of smoothness, and the historical decay further disrupts consistency when sub-instructions switch. This indicates the importance of incorporating a more global perspective into the value map utilizing. By clustering similar regions, the superpixel-based method enhances the understanding of the explored area, resulting in more stable navigation planning. This also explains why the superpixel-based method outperforms the Pixel-based approach.

**Generalization to other LLMs.** We further replace GPT-4 in our method with GPT-3.5 and Claude 3.5 Sonnet to explore the robustness and generalizability across different LLMs. From Table 6, we observe that Claude-3.5 Sonnet achieves a comparable SR to GPT-4 and even surpasses it in NE and SPL. This demonstrates that our method can adapt effectively to other LLMs. However, the performance of GPT-3.5 is not satisfactory. The reason could be that GPT-3.5 extracts less precise constraints than GPT-4 and Claude-3.5.

## 4.4 QUANTITATIVE AND QUALITATIVE ANALYSIS

**Economic and Low-latency.** We evaluate CA-Nav's response speed and cost efficiency. By using CSM for minimizing GPT-4 calls, CA-Nav outperforms NavGPT-CE significantly. As shown in

**Instruction:** Go straight. Pass the stairs on the right and continue straight. When you get to the stairs going up pass those as well. Go into the room with the couches and then turn right. wait near the glass table with white chairs.

Figure 5: Navigation visualization based on superpixel value map waypoint selection method.

Figure 4, NavGPT-CE takes about **1.29** seconds per action and costs about **$0.85** per episode, while CA-Nav responds in **0.12** seconds and costs only **$0.04** per episode, making it roughly 10 times faster and 5% of the cost.

**Navigation Visualization.** We visualize the navigation process of CA-Nav, NavGPT-CE and FBE-Nav in Figure 5, Figure 9, and Figure 10, respectively. Note that the FBE-Nav uses frontier-based exploration for waypoint selection, while the rest of the process is the same as the CA-Nav. Among these three methods, CA-Nav shows the most stable and coherent navigation, thanks to CSM's effective instruction decomposition and constraint identification, along with CVM's comprehensive use of the value map. In contrast, FBE-Nav fails at step 56 due to abrupt value changes at the frontier. NavGPT-CE struggles with inaccurate waypoint predictions and imprecise scene descriptions, leading to failures. More analysis are in Appendix § A.3

## 4.5 Real Robot Experiments

We conduct real robot experiments based on the QiZhi mobile robot[3], as shown in Figure 6 (e). We equip the robot with a laptop (including an Intel i9-14900HX CPU and a GeForce RTX 4090 GPU) and a Kinect V2.0 RGB-D camera whose HFoV is 84° and VFoV is 42°. However, due to the camera's limited depth sensing range and the inaccuracies or depth loss at the edges, we utilize Depth Anything V2 (Yang et al., 2024) to generate depth images. An

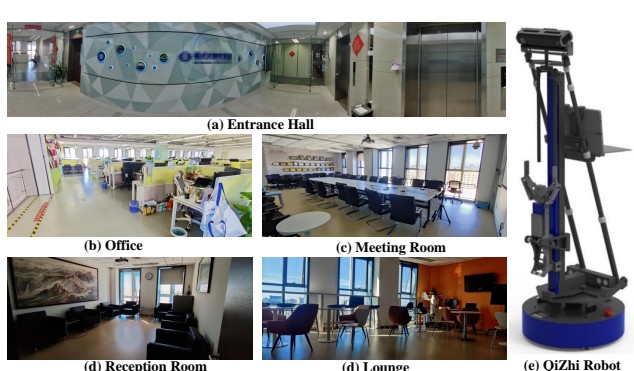

Figure 6: Real robot and real-world scenes.

RPLIDAR-A2M8 LiDAR is also used to obtain a relatively accurate pose through Hector SLAM. It is important to note that Hector SLAM is employed solely for estimating the camera's pose, not for constructing a pre-built map. The robot has a radius of 22.5cm and a height of 137cm.

Corresponding to the experiments in simulation (§ 3), we still set the low-level action as MOVE FORWARD (0.25m), TURN LEFT/RIGHT (30°), and STOP. The only modification is that successful navigation is defined as the robot stopping within 1 meter of the destination. For waypoint navigation, we continue to rely on the FMM approach, rather than adopting ROS navigation packages or other trained PointNav policies (Anderson et al., 2018a). The linear and angular velocities are set to 0.1m/s and 0.1rad/s respectively.

---

[3]QiZhi robot

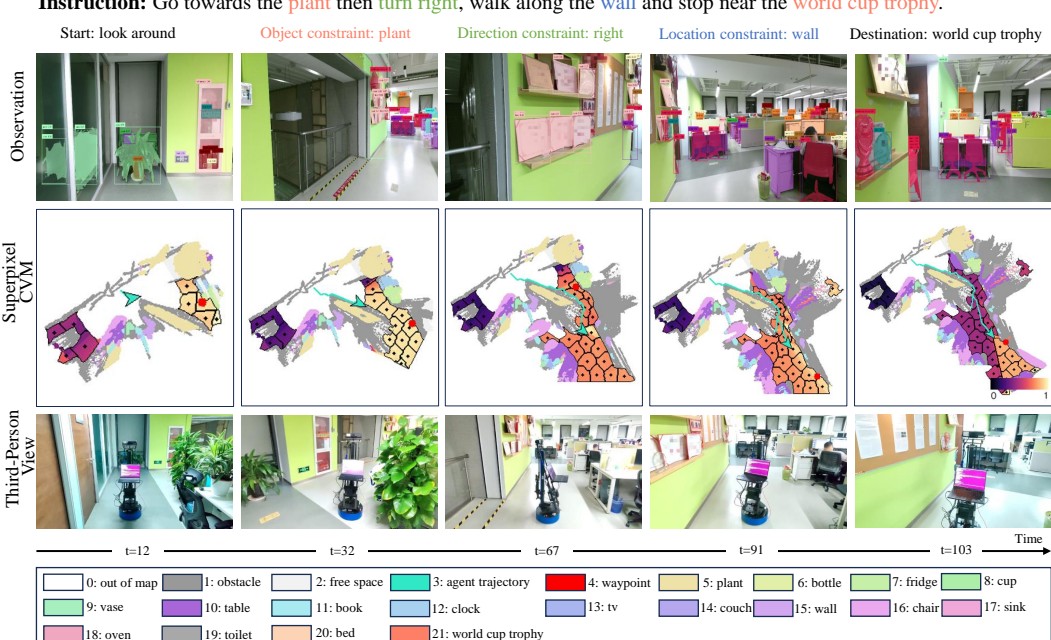

Figure 7: Visualization of the real-world navigation

To demonstrate the effectiveness of CA-Nav we conduct experiments in diverse indoor scenes including lounge, meeting room, reception room, entrance hall, and office, as shown in Figure 6. For instructions, we design 8 instructions with increasing complexity. Easy instructions contain only one sub-instruction and the destination is obvious, such as, "Go to the door.". Complex instructions are longer and with more than three constraints and the agent can not directly see the destination from its initial position, thus requiring exploration following the instruction. To check our method's ability to novel landmarks we design open vocabulary destinations such as "robot" and "world cup trophy" (details in Table 7).

For each instruction, we run 10 episodes, with the robot's initial pose slightly different each time (details in Table 7). The results indicate that even without a pre-built map, our real-time constructed constraint-aware value map is capable of handling long-horizon navigation tasks. Furthermore, thanks to the VLM, CA-Nav demonstrates a certain level of generalization to open vocabularies. As shown in Figure 7, the agent follows a complex instruction consisting of four sub-instructions and ultimately stops successfully near a World Cup Trophy. This indicates that the CA-Nav can generalize to new instructions and effectively track the navigation process.

## 5 CONCLUSION

In this work, we focus on developing a novel Constraint-Aware Navigator for the challenging zero-shot Vision-Language Navigation in Continuous Environments. To reach this goal we propose a Constraint-aware Sub-instruction Manager and a Constraint-aware Value Mapper. The two modules work coherently to navigate novel environments by identifying and adapting to the constraints of each sub-instruction. Experiments are conducted in both simulated and real-world environments. Our method not only outperforms other zero-shot methods in simulations but also demonstrates effectiveness in the real-world scenes. The current CA-Nav relies on closed-source LLMs, and the constraint-aware value map is affected by sensor noise. In the future, we aim to develop an approach that utilizes open-source LLMs and VLMs, relying solely on egocentric RGB cameras.

### ETHICS STATEMENT

This research develops an approach for zero-shot vision-language navigation, leveraging large language models and vision-language models to enhance autonomous navigation in indoor environments. While our approach has potential, it is crucial to acknowledge privacy concerns related to the use of open-source vision language models for object detection, as they may inadvertently capture sensitive information. Experiments were conducted in real-world settings with unavoidable

human presence. All participants, including scene owners and individuals in the laboratory, were informed about the study. They also provided their consent to participate. Additionally, our current implementation requires manual intervention to stop the robot, as it cannot autonomously detect and respond to dangers. This limitation poses safety risks in home environments. Addressing these ethical considerations is essential for the responsible use of this approach in diverse applications.

## REPRODUCIBILITY STATEMENT

We thoroughly explain our approach's architectures and implementation details in § 3. We describe our experimental setup in § 4.1 and LLM prompts in Appendix A.1. The pseudocodes are also provided in Appendix A.2 and the code can be found on our project page.

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

# A APPENDIX

## A.1 CA-NAV LLM PROMPT.

*[TASK DESCRIPTION]*
Parse a navigation instruction delimited by triple quotes and your task is to perform the following actions:
1. Extract Destination: Understand the entire instruction and summarize a description of the destination. The description should be a sentence containing landmark and room type. The description of the destination should not accurately describe the orientation and order. Here are examples about destination: "second room on the left" -> "room"(neglect order and direction); "between the bottom of the first stair and the console table in the entry way" -> "console table near entry way"(simplify description); "in front of the railing about halfway between the two upstairs rooms" -> "railing near two upstair rooms"
2. Split instructions: Split the instruction into a series of sub-instructions according to the execution steps. Each sub-instruction contain one landmark.
3. Infer agent's state constraints: Infer the state constraints that the agent should satisfy for each sub-instruction. There're thee constraint types: location constraints, direction constraints and object constraints. You need to select an appropriate constraint type and give the corresponding constraint object. Direction constraint object has two types: left, right. Constraints can format as a tuple: (constraint type, constraint object)
4. Make a decision: Analyze the landmarks, actions, and directions in each sub-instruction to determine how the agent should act. For a landmark, the agent has three options: approach, move away, or approach and then move away. For direction, the agent has three options: turn left, turn right, or go forward

*[OUTPUT DEFINITION]*
Provide your answer in JSON format with the following details:
1. use the following keys: destination, sub-instructions, state-constraints, decisions
2. the value of destination is a string
3. the value of sub-instructions is a list of all sub-instructions
4. the value of state-constraints is a JSON. The key is index start from zero and the value is a list of all constraints, each constraint is a tuple
5. the value of decisions is a nested JSON. The first level JSON's key is index start from zero and it's value is second level JONS with keys: landmarks, directions. The value of landmarks is a list of tuples, each tuple contains (landmark, action). The value of directions is a list of direction choice for each sub-instruction.

*[FEW-SHOT PROMPT]*
An Example:
User: "Walk into the living room and keep walking straight past the living room. Then walk into the entrance under the balcony. Wait in the entrance to the other room."
You: {{"destination": "entrance to the other room under the balcony", "sub-instructions": ["Walk into the living room", "keep walking straight past the living room", "walk into the entrance under the balcony", "wait in the entrance to the other room"], "state-constraints": {{"0": [["location constraint", "living room"]], "1": [["location constraint", "living room"]], "2": [["location constraint", "balcony"], ["object constraint", "entrance"]], "3": [["location constraint", "other room"], ["object constraint", "entrance"]]}}, "decisions": {{"0": {{"landmarks": [["living room", "approach"]], "directions": ["forward"]}}, "1": {{"landmarks": [["living room", "move away"]], "directions": ["forward"]}}, "2": {{"landmarks": [["balcony", "approach"], ["entrance", "approach"]], "directions": ["forward"]}}, "3": {{"landmarks": [["other room", "approach"], ["entrance", "approach"]], "directions": ["forward"]}}}}}}

*[KEY CONTENT REMINDER]*
ATTENTION:
1. constraint type: location constraint is for room type, object constraint is for object type, directions constraint. Don't confuse object constraint with location constraint!
2. landmark choice: approach, move away, approach then move away
3. direction choice: left, right, forward
4. The landmark and constraint object should not accurately describe the orientation and order. Here are examples about landmark: "second step from the top" -> "step"(neglect order and position relation); "room directly ahead" -> "room"; "right bedroom door" -> "bedroom door"

Figure 8: Instruction decomposition prompt.

Figure 8 illustrates the prompt details of CA-Nav. It consists of four parts, namely task description, output definition, few-shot prompt, and key content reminder. We find that it's helpful to give large language models an example to follow. Because the few-shot prompt can set a clear expectation of the desired output.

## A.2 PSEUDOCODE FOR THE CA-NAV

The full navigation process is detailed in Algorithm 1, with the $Check\_Constraints$ procedure explained further in Algorithm 2. Before navigation begins, an LLM provides a queue of constraints along with a description of the destination. During navigation, the Constraint-aware Sub-instruction Manager (CSM) keeps monitoring the process. At each step, CSM verifies whether the current set of constraints has been satisfied and determines if it needs to switch to the next sub-instruction.

**Algorithm 1** Navigation algorithm in an episode

1: **Input:** Instruction $\mathbf{I}$
2:        LLM Prompt $\mathbf{P}$
3: **Initialize:** Step Number $t \leftarrow 0$
4:           Agent Pose $\mathcal{P}_0 \leftarrow \varnothing$
5:           Constraints Queue $\mathcal{C} \leftarrow \varnothing$
6:           switch sub-instruction flag $\mathbf{S} \leftarrow False$
7:           search and go to destination flag $\mathbf{D} \leftarrow False$
8:           value map $\mathcal{V}_0 \leftarrow \mathbf{0}_{m \times m}$
9:           semantic map $\mathcal{M}_0 \leftarrow \mathbf{0}_{m \times m}$
10:          trajectory mask $\mathcal{T}_0 \leftarrow \mathbf{1}_{m \times m}$
11:
12: $\mathcal{C}, d \leftarrow Parse\_Instruction(\mathbf{P}, \mathbf{I})$        $\triangleright$ $d$ is the destination description extract by LLM
13: $c_t \leftarrow Get\_CurrentConstraints(\mathcal{C})$        $\triangleright$ get current set of constraints
14: $p_t \leftarrow Get\_LandmarkPrompt(c_t)$        $\triangleright$ get landmark prompt for value map generation
15: $\mathcal{O}_t, \mathcal{P}_t, \mathcal{M}_t \leftarrow LookAround()$
16: $\mathcal{T}_t \leftarrow Update\_TrajectoryMask(\mathcal{P}_t, \lambda)$
17: **while** Episode is not done **do**
18:     **if** $\mathbf{S}$ = True **and** $\mathcal{C}$ is not empty **then**        $\triangleright$ switch to next sub-instruction
19:         $c_t \leftarrow Get\_CurrentConstraints(\mathcal{C})$
20:         **if** length of $\mathcal{C} \leq 1$ **then**        $\triangleright$ reach the last sub-instruction
21:            $\mathbf{D} \leftarrow True$
22:            $p_t \leftarrow d$
23:         **else**
24:            $p_t \leftarrow Get\_LandmarkPrompt(c_t)$
25:         $\mathbf{S} \leftarrow False$
26:     $check \leftarrow Check\_Constraints(c_t, \mathcal{O}_t, \mathcal{P}_t)$        $\triangleright$ details in Algorithm2
27:     **if** all constraints in $c_t$ are checked as True **then**
28:         $pop(\mathcal{C})$
29:         $\mathbf{S} \leftarrow True$        $\triangleright$ ready to switch to next sub-instruction
30:     **else**
31:         $c_t \leftarrow Remove\_CheckedConstraints(check, c_t)$    $\triangleright$ only keep unsatisfied constraints
32:         $new\_p_t \leftarrow Get\_LandmarkPrompt(c_t)$
33:         **if** $new\_p_t \neq p_t$ **then**
34:            $\mathcal{V}_t \leftarrow \gamma \cdot \mathcal{V}_{t-1}$        $\triangleright$ use historical decay when switching sub-instruction
35:            $p_t \leftarrow new\_p_t$
36:     $v_t \leftarrow BLIP2(\mathcal{O}_t, p_t)$
37:     $\mathcal{V}_t \leftarrow Update\_ValueMap(\mathcal{M}_t, v_t, \mathcal{P}_t)$
38:     $\mathcal{V}_t \leftarrow \mathcal{T}_t \odot \mathcal{V}_t$        $\triangleright$ use trajectory mask to encourage exploration
39:     **if** $\mathbf{D}$ = True **then**
40:         $detection \leftarrow GroundingDINO(\mathcal{O}_t, d)$
41:         **if** $detection$ is not None **then**
42:            $w_t \leftarrow Project\_Location(detection, \mathcal{P}_t, \mathcal{M}_t)$
43:         **else**
44:            $w_t \leftarrow \pi(\mathcal{V}_t)$
45:     **else**
46:         $w_t \leftarrow \pi(\mathcal{V}_t)$
47:     $a_t \leftarrow FMM(w_t, \mathcal{M}_t)$        $\triangleright$ plan low-level action based on the waypoint and map
48:     $t \leftarrow t + 1$
49:     $\mathcal{O}_t, \mathcal{P}_t, done \leftarrow Execute\ a_t$
50:     **if** done is True **then**
51:         break
52:     $\mathcal{T}_t \leftarrow Update\_TrajectoryMask(\mathcal{P}_t, \lambda)$
53:     $\mathcal{M}_t \leftarrow Update\_SemanticMap(\mathcal{O}_t, \mathcal{M}_t, \mathcal{P}_t)$
54: **Result:** Episode ends.

It's worth noting that the current set of constraints $c_t$ may include multiple constraints, which will be checked individually as shown in Algorithm 2. Specifically, the function $Check\_Constraints$ identifies each constraint $c_i$ in $c_t$, applies the corresponding method to check it and stores the results in a list. Next, CSM will pop the current element from the queue if all constraints in $c_t$ are satisfied. Otherwise, it will remove the constraints that have already been met and check the remaining constraints in the next step. When the prompt provided by $c_t$ changes, the value map built from the previous prompt will be decayed by a factor of $\gamma$. As the agent navigates, the trajectory mask $\mathcal{T}_t$ tracks the agent's trajectory and reduces the willingness to explore within those areas. This mask is a 2D matrix, initialized to all ones and matching the shape of the map. Similar to historical decay, the trajectory mask decays by a factor $\lambda$.

---

**Algorithm 2** Check Constraints algorithm

---

1: **Input:** Current set of constraints $c_t$
2:        Current observation $\mathcal{O}_t$
3:        Current agent pose $\mathcal{P}_t$
4: **Output:** Checklist of current set of constraints $check$
5: **Initialize:** Constraints check results $checks \leftarrow \varnothing$
6:
7: **for** $c_i$ in $c_t$ **do**
8:     **if** $c_i$ is object constraint **then**
9:         $checks \leftarrow chekcs \cup \{check\_object\_constraint(c_i, \mathcal{O}_t)\}$     ▷ utilize Grounding DINO
10:     **if** $c_i$ is location constraint **then**
11:         $checks \leftarrow chekcs \cup \{check\_location\_constraint(c_i, \mathcal{O}_t)\}$          ▷ utilize BLIP2 VQA
12:     **if** $c_i$ is direction constraint **then**
13:         $checks \leftarrow chekcs \cup \{check\_direction\_constraint(c_i, \mathcal{P}_t)\}$

---

### A.3 MORE ANALYSIS OF VISUALIZATION.

**Analysis of NavGPT-CE.** As shown in Figure 9, the agent initially follows the instructions correctly, but errors start to appear from step 30. There are two main types of errors: The first type is due to the waypoint model failing to accurately predict navigable nodes, as seen in step 30. At step 30, the ground truth action was to move forward, and the LLM correctly identified this. However, the waypoint model failed to predict the navigable viewpoint, causing the agent to move to the right instead. The second type arises from the caption model incorrectly describing the scene, as illustrated in steps 41, 55, and 125. At step 55, the caption model describes the current panoramic observation as "a bathroom with a toilet and a sink." However, the primary objects in the scene are stairs, and this incorrect caption ultimately misleads the agent. Our analysis indicates that NavGPT-CE converts all visual observations into text and makes navigation decisions based solely on scene descriptions and a summary of the history. This approach can easily lead to incorrect decisions when the descriptions are inaccurate.

**Analysis of FBE-based waypoint selection method.** As shown in Figure 10, the agent navigates correctly during the first sub-instruction. However, upon reaching the second sub-instruction, the landmark prompt changes to "room," causing the value map, initially based on the previous landmark prompt "stairs", to decay as new values are updated. The FBE-based waypoint selection method then chooses a left frontier, which has the highest value among the available options, but the correct path is to walk straight toward the open area with the stairs. Compared to the Superpixel Value Map based waypoint selection method shown in Figure 5, CA-Nav avoids focusing solely on local high values. Instead, it considers the global value map, allowing it to navigate correctly during the second sub-instruction.

**Instruction: Go straight. Pass the stairs on the right and continue straight. When you get to the stairs going up pass those as well. Go into the room with the couches and then turn right. wait near the glass table with white chairs.**

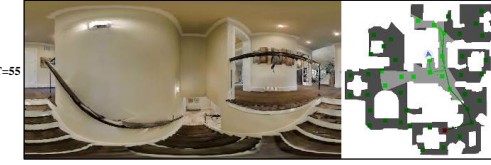

**T=1**

**Observation:**

Current Viewpoint "2e880bca51f7e5b920f8e3719638b2c0":

…

Direction: Rear Left, range (left 157.50 to left 112.50):
**Caption: 'a hallway with a staircase leading to a living room'**
Rear Left Objects in 3m: None
Rear Left Navigable
Viewpoints:{'96b3728375ec14aa18a7a36bb378bcab': 'left 156.00, 1.25m'}

…

**Thought:** From the current viewpoint, **I can see a hallway with a staircase leading to a living room to the rear left.** This aligns with the instruction to pass the stairs on the right. Therefore, I should move to the viewpoint in the rear left direction ('96b3728375ec14aa18a7a36bb378bcab') to continue following the instruction.
**Action:** "96b3728375ec14aa18a7a36bb378bcab"

**T=30**

**Observation:**

Current Viewpoint "ff1a2e47bef979a2cd0b063adfce2f5d":

…

Direction: Right, range (right 67.50 to right 112.50):
**Caption: 'a house with a doorway and a table and chairs'**
Right Objects in 3m: None
Right Navigable
Viewpoints:{'8c69af1b139dac31f9a25b3d9a79aced': 'right 72.00, 1.25m'}

…

**Thought:** From the current viewpoint, I can see a hallway with a white door and a painting on the wall to the front right, and a house with a doorway and a table and chairs to the right. **The instruction tells me to go into the room with the couches and then turn right. Since there are no navigable viewpoints directly in front of me, I should move to the right** viewpoint ('8c69af1b139dac31f9a25b3d9a79aced').
**Action:** "8c69af1b139dac31f9a25b3d9a79aced"

**T=41**

**Observation:**

Current Viewpoint "8c69af1b139dac31f9a25b3d9a79aced":

…

Direction: Right, range (right 67.50 to right 112.50):
**Caption: 'a bathroom with a toilet and a sink in it'**
Right Objects in 3m: None
Right Navigable
Viewpoints:{'8077400fd43c0c5d4dc68c3fd011b3b3': 'right 90.00, 1.00m'}

…

**Thought:** From the current viewpoint, I can see a dining room with a table and chairs in front of me, which aligns with the instruction to go into the room with the couches. **The instruction also tells me to turn right and wait near the glass table with white chairs. Since there is a navigable viewpoint to the right ('8077400fd43c0c5d4dc68c3fd011b3b3'), I should move to that viewpoint** to continue following the instruction.
**Action:** "8077400fd43c0c5d4dc68c3fd011b3b3"

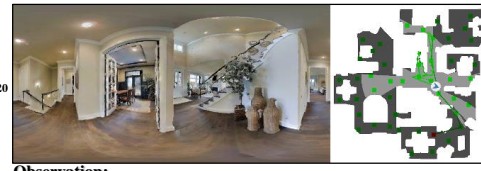

**T=55**

**Observation:**

Current Viewpoint "2e880bca51f7e5b920f8e3719638b2c0":

…

Direction: Front, range (left 22.50 to right 22.50):
**Caption: 'a bathroom with a toilet and a sink'**
Front Objects in 3m: None
Front Navigable Viewpoints:{'5edaddee63e3a454aa645602ee8cb89f': 'right 9.00, 1.00m'}

…

**Thought:** From the current viewpoint, I can see a bathroom with a toilet and a sink in front of me, and a living room with a fireplace in the rear right. The instruction tells me to go into the room with the couches and then turn right, waiting near the glass table with white chairs. **Since there is no glass table with white chairs visible from this viewpoint, I should move to the next navigable viewpoint that might lead me closer to the target**. The closest navigable viewpoint is to the front ('5edaddee63e3a454aa645602ee8cb89f').
**Action:** "5edaddee63e3a454aa645602ee8cb89f"

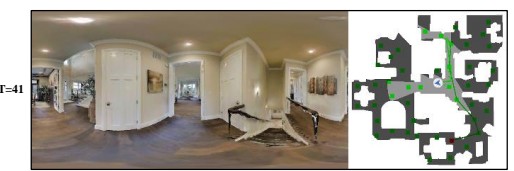

**T=120**

**Observation:**

Current Viewpoint "96f06c10564f541994a4cbe12c94e8ee":

…

Direction: Left, range (left 112.50 to left 67.50):
**Caption: 'a glass door in a room with a table and chairs'**
Left Objects in 3m: None
Left Navigable Viewpoints:{'cdf8ef16efbf982bdbe434b2b1933608': 'left 84.00, 1.00m'}

…

**Thought:** From the current viewpoint, I can see a living room with a couch and a fireplace in front of me, and a living room filled with furniture and a fireplace to the front right. The instruction tells me to go into the room with the couches and then turn right, waiting near the glass table with white chairs.
Given the current layout, **the left navigable viewpoint seems to lead to a room with a table and chairs**, which might be closer to the target location described in the instruction. I should move to that viewpoint to continue following the instruction.
**Action:** "cdf8ef16efbf982bdbe434b2b1933608"

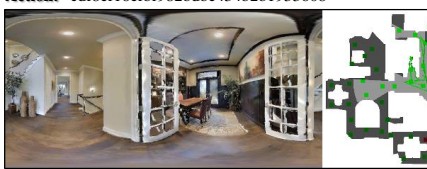

**T=135**

**Thought:** I have now reached a viewpoint where I can see a dining room with a table and chairs in front of me, which matches the instruction to wait near the glass table with white chairs. Since the instruction specifies to **wait near the glass table with white chairs** and I am within 3 meters of it, I have reached the destination.

**Final Answer: Finished!**

**Legend:**

**Bold Green font with underline**: correct choice and reasoning

**Bold Red font with underline**: wrong choice and reasoning

Figure 9: Case study for NavGPT-CE.

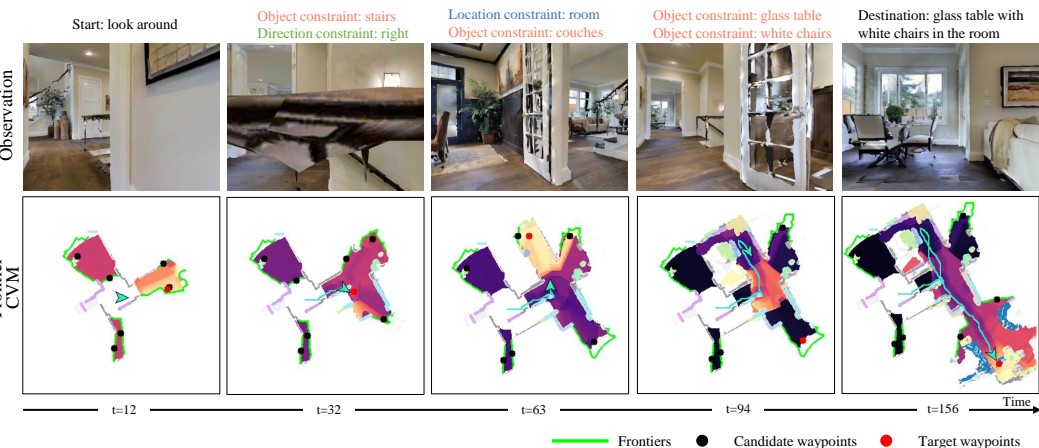

Figure 10: Navigation visualization based on FBE-based waypoint selection method

Table 7: Information about real robot experiments. The robot's initial pose will be slightly different in each episode. $d$ represents the distances between the initial position and the destination. SR is the success rate.

| Instruction | Initial Pose | $d$ | SR |
|---|---|---|---|
| Go to the door. |  | 5.4m | 8/10 |
| Turn left then walk towards the signboard and wait by the elevator. |  | 5.5m | 4/10 |
| Walk out of the door then stop in front of the plant. |  | 6.6m | 5/10 |
| Walk out to the corridor, and stop in front of the poster. |  | 4.3m | 6/10 |
| Walk towards the robot. |  | 3.6m | 7/10 |
| Walk towards the living room then stop beside the couch |  | 6.0m | 4/10 |
| Turn slightly right and walk into the meeting room, step forward then stop in front of the table. |  | 3.8m | 4/10 |
| Go towards the plant, turn right, walk along the wall then stop near the World Cup Trophy. |  | 10.4m | 2/10 |

