# OpenReview forum: "CONSTRAINT-AWARE ZERO-SHOT VISION-LANGUAGE NAVIGATION IN CONTINUOUS ENVIRONMENTS"
_ICLR.cc/2025/Conference — ICLR 2025 Conference Withdrawn Submission_

### Official Review · Reviewer_GVTf · 2024-10-16

**Soundness:** 3
**Presentation:** 3
**Contribution:** 1
**Rating:** 3
**Confidence:** 4

**Summary:**

This paper presents an LLM-based visual navigation agent CA-Nav, which is composed mainly of three parts - CSM, CVM, and Waypoint Selection Module. The CSM is a task-level planning module that decomposes the language instructions into sub-instructions.  Based on the sub-instructions, the CVM assigns the affordance values on the map, and finally, the Waypoint Selection Module translates the value maps into a collision-free navigation trajectory.  Foundation models such as GroundingDINO, SAM, and BLIP2 are used to provide a comprehensive environment perception. The proposed approach is fairly evaluated in the VLN-CE tasks, including both R2R-CE and RxR-CE, and achieves better performance with several zero-shot VLN-CE baseline methods. Real-world experiments are also conducted to support the effectiveness of the proposed method.

**Strengths:**

(1) This paper proposes a zero-shot LLM Agent for visual-language navigation in a continuous environment. Both the simulation results and the real-world videos prove the effectiveness of the proposed approach.

(2) This paper proposes a novel Superpixel-based Waypoint Selection module, which is able to deal with the inconsistency values in generating the value maps and generates a smoother navigation trajectory.

(3) The detailed ablation study and analysis help understand the function of each proposed approach.

(4) The real-world deployment shows the potential of the proposed approach.

**Weaknesses:**

(1) The technique contribution is limited. Using the LLM to decompose long-horizon navigation tasks into sub-instructions has been proposed in the work InstructNav[1], which also uses landmark, action as constraints for sub-tasks. Using BLIP-2 for generating the value map has been proposed in the paper VLFM[2]. It seems that the only difference is that VLFM uses the object goal as the language query but CA-Nav uses a decomposed instruction as the query.

(2) The overall performance on the VLN-CE benchmarks is not satisfying. The CA-Nav only got ~25% success rate which is relative lower than the recent LLM-based approaches such as InstructNav and Navid[3].

(3) The system is rather complicated and introduces multiple foundation models during the decision process, including . The decision frequency/speed is an important issue for robotics applications.

[1] Long, Yuxing, et al. "InstructNav: Zero-shot System for Generic Instruction Navigation in Unexplored Environment." arXiv preprint arXiv:2406.04882 (2024).

[2] Yokoyama, Naoki, et al. "Vlfm: Vision-language frontier maps for zero-shot semantic navigation." 2024 IEEE International Conference on Robotics and Automation (ICRA). IEEE, 2024.

[3] Zhang, Jiazhao, et al. "NaVid: Video-based VLM Plans the Next Step for Vision-and-Language Navigation." arXiv preprint arXiv:2402.15852 (2024).

**Questions:**

(1) As mentioned in Section 3.2, there are three types of constraints including landmark, location, and action. Only when the constraints are satisfied or exceed the time limit, the next sub-instructions will be used for the next decision. However, it seems that in the subsequent steps such as value map generation and path planning, there are no explicit schemes to guarantee the navigation process will fulfill the constraints. The constraint attributes are only used for querying the BLIP-2. Does the BLIP-2 score enough to finish the sub-tasks and walk a perfect trajectory to solve all the constraints?

(2) There are clear pre-defined rules for deciding whether the sub-instructions are finished. But what if your agent mistakenly arrives at a proper next waypoint without satisfying the first sub-instructions, should it go back and try to complete the first step or directly skip the first and continue for the rest of the sub-tasks? Is there a more flexible framework for such scenarios in your method?

(3) How long does it take for the whole round of the planning process, including semantic mapping, generating sub-instructions, value maps, and path planning?

(4) Why does the success rate vary for the same instruction and same starting positions shown in the supplementary Table 7? What influences the performance and makes the approach sometimes can finish the task and sometimes can not.

---

> ### Author Response · Authors · 2024-11-21
> **Response to Reviewer GVTf (Questions)**
>
> Thank you very much for your review and comments.
>
> ### Q1: There are no explicit schemes to guarantee the navigation process will fulfill the constraints. Does the BLIP-2 score enough to finish the sub-tasks and walk a perfect trajectory to solve all the constraints?
>
> Thank you for raising this concern. BLIP-2 is a suitable choice for our task, as demonstrated by its success in VLFM[1], where it effectively handles open-vocabulary object queries for object navigation. While it is difficult to guarantee that all constraints are fully satisfied due to the inherent imperfections of current perception models, we have designed two complementary mechanisms to enhance constraint fulfillment:
> 1. Trajectory Mask: This mechanism encourages the agent to explore new areas, improving the chances of satisfying constraints.
> 2. Historical Decay: This mitigates abrupt changes in the value map when switching sub-instructions, providing smoother navigation.
>
> [1] Yokoyama, Naoki, et al. "Vlfm: Vision-language frontier maps for zero-shot semantic navigation." 2024 IEEE International Conference on Robotics and Automation (ICRA). IEEE, 2024.
>
> ### Q2: Will the agent always inflexibly go back and try to complete the first step when mistakenly arrives at a proper next waypoint without satisfying the first sub-instructions? Is there a more flexible framework for such scenarios in your method?
>
> We didn't design a specific error correction module. However, we did experiments in both simulation and real-world and found that it rarely goes back. On one hand, we have the trajectory mask to discourage the agent from going back to the explored area. On the other hand, as long as the sub-instruction switches, historical decay would give the past value map a low weight, which helps the agent flexibly skip the previous hard constraints and concentrate on current constraints.
>
> ### Q3: How long does it take for the whole round of the planning process, including semantic mapping, generating sub-instructions, value maps, and path planning?
>
> We studied the model efficiency in section 4.4 and we reported that the proposed CA-Nav took about 0.12 seconds for each action decision. One action decision procedure (i.e. the planning process) includes: semantic mapping, value mapping, and FMM path planning. Processing one instruction took about 13s but we didn't take this part into account since it would be done before the episode started.
>
> ###　Q4: Why does the success rate vary for the same instruction and same starting positions shown in the supplementary Table 7?
>
> The variability in success rate primarily stems from the robot's vibration during navigation, which can cause fluctuations in Lidar data. These variations affect camera pose estimation, leading to slight differences in the dynamically constructed map and subsequent waypoint selection.

---

> > ### Author Response · Authors · 2024-11-21
> > **Response to Reviewer GVTf (Weakness)**
> >
> > ### Q1: The contribution is limited, especially when compared to InstructNav (for sub-instruction decomposition with constraints) and VLFM (for using BLIP-2 to generate value maps).
> >
> > Thank you for highlighting the comparison with InstructNav and VLFM. We will cite these works in the paper and discuss them accordingly. Below, we provide a holistic comparison to clarify how CA-Nav differs from these approaches:
> >
> > 1. Observation Space:
> > InstructNav relies on panoramic observations, which provide a broader field of view but are less realistic for deployment on standard robots with egocentric sensors. In contrast, CA-Nav operates with egocentric observations, aligning better with real-world robotics hardware constraints. Similarly, VLFM focuses on frontier-based exploration, limiting its decision-making to the boundary of the explored area. CA-Nav utilizes the entire value map, providing a larger and more flexible decision space.
> >
> > 2. Sub-task Switching and Navigation Mechanism:
> > CA-Nav introduces a constraint-based instruction switching mechanism, where sub-task transitions are predefined based on explicit state constraints, such as landmarks, locations, and directions. This approach eliminates the need for real-time LLM calls during navigation, ensuring faster (0.12s per decision-making, in Figure 4) and more cost-effective (0.04$ per trajectory, in Figure 4) decision-making. In contrast, InstructNav requires invoking GPT-4 at each decision step to re-evaluate the navigation chain, which, while effective, can lead to higher computational costs and latency.
> >
> > 3. Task Complexity:
> > VLFM focuses on object navigation, where the agent's goal is simply to locate specific objects. CA-Nav addresses the more complex Vision-Language Navigation in Continuous Environments (VLN-CE) task, requiring the agent to follow natural language instructions, manage sequential sub-tasks, and track navigation progress. This makes CA-Nav suitable for long-horizon, instruction-driven navigation tasks.
> >
> > 4. System Efficiency and Stability:
> > CA-Nav incorporates trajectory masking (improved 10% sr, Table 4) and historical decay (improved 10% sr, Table 4) to refine the value map, enhancing navigation stability and mitigating abrupt changes during sub-task switching. These techniques, combined with the superpixel clustering method (improved 15.5% sr, Table 5) for waypoint selection, contribute to CA-Nav's robust and efficient decision-making process, as evidenced by the improvements shown in Table 5.
> >
> > By addressing these four aspects comprehensively, CA-Nav demonstrates its distinctiveness and contributions beyond InstructNav and VLFM. Moreover, InstructNav and CA-Nav are concurrent works.
> >
> > ### Q2: The overall performance on the VLN-CE benchmarks is not satisfying. The CA-Nav only got ~25% success rate, which is relatively lower than the recent LLM-based approaches such as InstructNav and Navid.
> >
> > - Comparison with InstructNav:
> >   1. Cost Efficiency: InstructNav relies on closed-source multimodal models like GPT-4V, which are costly to use and inaccessible for many applications. CA-Nav utilizes open-source models, making it more economical and practical. Notably, InstructNav's ablation study shows that using open-source MLMs (e.g., LLaVA sr=17) results in lower performance than CA-Nav (sr=25.3).
> >   2. Challenging Setup: InstructNav leverages panoramic observations and GPT-4V for decision-making. While CA-Nav focuses on a more realistic and challenging setting that only uses egocentric observation.
> > - Comparison with Navid:
> >   1. The experimental setting of CA-Nav is different from Navid. Navid fine-tunes LLMs using the R2R dataset, leveraging additional training data to improve performance. In contrast, CA-Nav is entirely training-free, relying solely on foundation models.

---

> > > ### Comment · Reviewer_GVTf · 2024-11-21
> > > **Questions about the CA-Nav scalability to dynamic scenes**
> > >
> > > Thank you for your reply to my previous questions. I think it make sense that the CA-Nav can serve as a more efficient and economic navigation method compared with InstructNav and the CA-Nav deal with the VLN-CE instead of the ObjectNav compared with VLFM. But I still have the following questions:
> > >
> > > (1) Based on your answers to the Q2 about error correction module, if the sub-tasks are decomposed in advance, how to deal with the scenarios with the similar objects' existance. For example, if there are two doors in front of the robot, "enter the door" can be one of the decomposed sub-tasks from LLM, once the robot enters the wrong door, it cannot come back and correct its behavior.
> > >
> > > (2) Similar to (1), without online task planning process, dynamic obstacles (such as the movement of the person) can easily interrupt the original plan. Can CA-Nav deal with such scenarios?
> > >
> > > (3) I am still curious about the experiment result variance in Q4. How significantly does the vibration of the robot influence the estimated camera pose?  Why the robot running with a slow speed will introduce such a huge mapping variance with the lidar inputs?

---

> > > > ### Author Response · Authors · 2024-11-23
> > > > **Response to Reviewer GVTf (Questions about the CA-Nav scalability to dynamic scenes, Q1 and Q2)**
> > > >
> > > > Thank you very much for your timely response and for acknowledging our previous answers! Once again, we appreciate the questions you have raised. Below are our responses:
> > > >
> > > > ### Q1: If the sub-tasks are decomposed in advance, how to deal with the scenarios with similar objects' existence?
> > > > Thank you for your thoughtful example. To address this scenario, we would like to first emphasize that VLN tasks, unlike Object Navigation, typically include descriptive instructions to differentiate between similar objects. These descriptions often specify attributes such as appearance, location, or orientation, providing clear guidance to the agent.
> > > > In your example, where two doors are present, the VLN instruction would typically specify which door to approach. For instance, in R2R-CE dataset, one instruction (id: 1470) is:
> > > >
> > > > "Go through the right doorway, and wait near the top of the stairs going down.".
> > > >
> > > > The instruction decomposition results are:
> > > > "1470": {
> > > >         "destination": "top of the stairs",
> > > >         "sub-instructions": ["Go through the right doorway","wait near the top of the stairs going down"],
> > > >         "state-constraints": {
> > > >             "0": [["direction constraint","right"],["object constraint","doorway"]],
> > > >             "1": [["location constraint","stairs"]]}
> > > > }
> > > >
> > > > Due to the instruction containing orientation description (i.e. "right"), CA-Nav can identify the direction constraint. As shown in the navigation video (video: <https://youtu.be/1dQMidn_hLU>), CA-Nav’s constraint detection module effectively interprets such distinctions, enabling the robot to identify and choose the correct door. This demonstrates the system’s robustness in scenarios where clear instructions are provided.
> > > >
> > > > We agree that incorporating a correction mechanism to handle such a scenario is an important research direction. CA-Nav can handle such issues to some extent based on constraints (e.g., ["direction constraint", "right"]). However, we acknowledge that CA-Nav cannot fully resolve this and it is a common challenge in current zero-shot VLN-CE task. We plan to address this in future work.
> > > >
> > > > ### Q2: Without an online task planning process, dynamic obstacles (such as the movement of the person) can easily interrupt the original plan. Can CA-Nav deal with such scenarios?
> > > >
> > > > CA-Nav is capable of handling scenarios with sparse crowds, as demonstrated in our real-world experiments (video: <https://youtu.be/0Y8bZxwzJ3s>). In this experiment, we encountered situations where moving persons became obstacles in the map. However, as long as there was sufficient free space remaining for the robot to navigate, CA-Nav was able to find an alternative path and successfully complete the task.
> > > > Moreover, we agree that navigation in dynamic environments is indeed an interesting and critical direction. Our method is not fully developed for highly dynamic environments. Handling this challenge may require advanced techniques such as 3D scene graphs[1], which we will explore in the future.
> > > >
> > > > [1] hijie Yan and Shufei Li and Zuoxu Wang and Lixiu Wu and Han Wang and Jun Zhu and Lijiang Chen and Jihong Liu. Dynamic Open-Vocabulary 3D Scene Graphs for Long-term Language-Guided Mobile Manipulation. arXiv preprint arXiv: 2410.11989

---

> > > > > ### Author Response · Authors · 2024-11-23
> > > > > **Response to Reviewer GVTf (Questions about the CA-Nav scalability to dynamic scenes, Q3)**
> > > > >
> > > > > ### Q3: How significantly does the vibration of the robot influence the estimated camera pose? Why the robot running with a slow speed will introduce such a huge mapping variance with the lidar inputs?
> > > > >
> > > > > Thanks for your careful review! Although the robot moves at a relatively slow speed, the acceleration during the start and stop phases can cause vibrations and the uneven ground also leads to vibrations. Even with slow robot speeds, small perturbations in sensor readings can propagate through the mapping and decision-making pipeline and finally causing the success rate variation. However, quantifying the specific impact of vibrations on the camera's pose is challenging. Instead, we use a simple spirit level to visually demonstrate the intensity of the vibrations (video: <https://youtu.be/VUXbWEbcil8>). The video demonstrates that the robot does experience noticeable vibrations during its movement.
> > > > >
> > > > > In addition, the robot's base has execution errors, such as imprecise steering, movement distances, and skid. These factors can cause disturbances to the robot, leading to the accumulation of errors. This means that even running a predefined trajectory several times, the robot may end in different positions. To clarify this, we conducted an additional experiment. We first fixed the robot's starting position and orientation, then let it follow a predefined trajectory consisting of 20 steps. The experiment was repeated 10 times, and the final pose of the robot was recorded each time (video: <https://youtu.be/9Mg9e72ErfY>). The results showed significant deviations between the stopping poses across different trials, even with such a short trajectory. This indicates that in our more complex real-world experiments, these accumulated errors may lead to variations in success rates.
> > > > >
> > > > > We plan to explore hardware-level improvements to alleviate this issue.

---

### Official Review · Reviewer_cEek · 2024-10-25

**Soundness:** 3
**Presentation:** 3
**Contribution:** 2
**Rating:** 6
**Confidence:** 3

**Summary:**

This paper addresses the challenging task of Vision-Language Navigation in Continuous Environments (VLN-CE). The task presents two main challenges: 1) Continuous Environments large state space 2) requires an understanding of visual observation details that are hard to summarize by language alone.

The key idea is to perform the Constraint-aware Sub-instruction Manager (CSM) and define the completion criteria of decomposed sub-instructions as constraints, and track navigation progress. Then, the Constraintaware Value Mapper (CVM) builds a value map that is continually updated using a constraint prompt and semantic map.

**Strengths:**

The overall idea of using a pre-trained vision language model for navigation has been extensively studied in the recent literature. However, this paper's approach seems well thought out. The constraint formulation (direction, object, and location), while straightforward, does make sense and seems effective in breaking down complex navigation tasks.

The superpixel clustering method for waypoint selection is a smart way to handle noisy and high dimensional value maps and allows continuous action space.

I like that the comprehensive ablation studies validate all key design decisions.

I also like the fact that the author provides real-world navigation results on real robots.

**Weaknesses:**

I'm not actively working in this field, and it is a bit hard for me to judge the technical novelty of the method.
However, the high-level ideas seem standard, despite the naming differences: It contains high-level planning (called CSM in the paper) and low-level policy (called CVM in the paper). Using constraints in high-level planning and using them to inform low-level policy is also a typical approach for task and motion planning, but this time with VLM and LLM to better handle unstructured images.  Reading the paper method seems to be a straightforward combination of existing models. The majority of the performance improvement probably comes from the pre-trained models.

**Questions:**

Question to the author:

- Is LLM really needed here? The construction of the constraints based on instruction seems a pretty structured task; a simple rule-based method or small language model could work just fine.

- The performance seems a bit low (25% success rate in HM3D, despite already being higher than some of the baselines compared to). It seems that the system is not working most of the time. This makes me think? 1) What are the common failure cases? 2) Is the task naturally ambiguous or ill-defined? Can humans do this task with the same observation space?

---

> ### Author Response · Authors · 2024-11-21
> **Response to Reviewer cEek**
>
> Thank you very much for your review and comments.
>
> ### Q1: Is LLM really needed here?
> 1. The paper CLIP-Nav[1] highlights the advantages of LLMs, particularly GPT-3, in processing complex instructions. LLMs excel at capturing the sequential structure of instructions and handling nuanced language, which would be challenging for rule-based approaches. Similarly, CA-Nav leverages LLMs to identify and interpret constraints effectively, including distinguishing between object and location constraints within the same sub-instruction. It's a task that is non-trivial for simpler methods.
> 2. Our ablation study in Table 6 demonstrates that stronger LLMs (e.g., GPT-4) significantly improve CA-Nav's performance compared to smaller models, indicating the benefits of using LLMs.
>
> ### Q2:  What are the common failure cases? Is the task naturally ambiguous or ill-defined? Can humans do this task with the same observation space?
>
> 1. The common failure cases in our approach can be categorized into two main types:
>     - Constraints Detection Errors: The agent may fail to correctly identify constraints, such as landmarks or directional cues, leading to incorrect switching decisions.
>     - Suboptimal Path Planning: Errors in waypoint selection often stem from inaccuracies in the value map generated from BLIP2. When the value map does not accurately represent the environment or constraints, the agent may select inefficient or incorrect paths.
> In summary, the main cause of failure lies in perception modules. We use several perception modules to detect constraints and build value maps. However, they can hardly be completely correct all the time. Moreover, the 3D reconstruction in Habitat Simulator has low visual fidelity and low physical fidelity which would futhrer affect the percetion mouldes. Similar discoveries and opinions are proposed in [2]. With the development of foundation models, these problems could be mitigated.
>
> 2. The task is neither ill-defined nor ambiguous. Wang et al.[3] shows that humans can achieve a success rate of approximately 0.86, and agents trained on large-scale data can approach human-level performance, indicating the task's well-defined structure and learnability. The primary challenge lies in the zero-shot setup, where the agent lacks prior knowledge of long-horizon navigation strategies and environmental understanding.
>
> [1] Vishnu Sashank Dorbala and Gunnar Sigurdsson and Robinson Piramuthu and Jesse Thomason and Gaurav S. Sukhatme. CLIP-Nav: Using CLIP for Zero-Shot Vision-and-Language Navigation. arXiv preprint arXiv: 2211.16649
>
> [2] Theophile Gervet and Soumith Chintala and Dhruv Batra and Jitendra Malik and Devendra Singh Chaplot. Navigating to Objects in the Real World. arXiv preprint arXiv: 2212.00922
>
> [3] Zun Wang and Jialu Li and Yicong Hong and Yi Wang and Qi Wu and Mohit Bansal and Stephen Gould and Hao Tan and Yu Qiao. Scaling Data Generation in Vision-and-Language Navigation. arXiv preprint arXiv: 2307.15644

---

### Official Review · Reviewer_xtPJ · 2024-11-04

**Soundness:** 3
**Presentation:** 3
**Contribution:** 2
**Rating:** 5
**Confidence:** 4

**Summary:**

This paper presents CA-Nav, a zero-shot method for Vision-Language Navigation in Continuous Environments (VLN-CE). It reframes VLN-CE as a sequential sub-instruction process, leveraging two modules: the Constraint-aware Sub-instruction Manager (CSM) and the Constraint-aware Value Mapper (CVM). CSM divides instructions into sub-instructions and tracks progress, while CVM builds value maps to enhance navigation stability. CA-Nav achieves sota results on R2R-CE and RxR-CE, and proves effective in real-world robot deployments.

**Strengths:**

1. The use of the Constraint-aware Sub-instruction Manager (CSM) to decompose instructions and generate constraints for landmarks, locations, or directions, and switch sub-instructions based on these constraints is an effective and simple mechanism that enhances the robustness of the navigation process.
2. Unlike simple map construction, the paper introduces the Constraint-aware Value Mapper (CVM), which calculates the similarity to constrained landmarks.
3. Performance: The method achieves state-of-the-art results on the R2R-CE and RxR-CE datasets, surpassing other methods in terms of success rate and success weighted by trajectory length (SPL). The paper also mentions that CA-Nav delivers 10x faster response times and 95% cost reduction compared to other approaches, which is crucial for real-world applications.
4. Real-world Application Potential: Tests in real-world robot deployments show that CA-Nav generalizes well across different environments and instructions, demonstrating its potential for practical use.

**Weaknesses:**

1. Lack of innovation: The CSM aims to leverage LLMs for instruction decomposition and track navigation progress through explicit sub-instruction switching, while the CVM constructs a constraint-based value map. However, these are existing approaches, including map building, value map, progress tracking, mileage information, landmarks, objects, and directions. The use of LLMs for decomposition and constraint extraction seems insufficient as a major point of innovation.

2. Limited benchmark scope: Although the paper performs well on two VLN-CE benchmark datasets, its generalization ability is unclear. It is recommended to evaluate the method on more diverse datasets or different embodied tasks to further verify its general applicability.

3. Lack of detail in instruction decomposition: Instruction decomposition is a key part of the system, but the paper does not provide enough details about the robustness of this process across different types of instructions. While the authors mention using a large language model (LLM), the handling of complex instructions has not been fully explored.

4. Over-reliance on constraints: Although the constraint-aware approach is innovative, over-reliance on constraints in some cases may lead to suboptimal behavior. For example, if the robot incorrectly detects a landmark or misinterprets a constraint, it might switch sub-instructions prematurely or delay switching. The paper does not delve into failure cases or limitations of the method.

5. Lack of some ablation studies: Despite strong experimental results, there is a lack of ablation studies to analyze the contribution of each module. For example, the impact of CSM, CVM, and superpixel clustering on overall performance has not been thoroughly analyzed. Such studies would help better understand the importance of each module.

**Questions:**

1. Difference from VLFM: VLFM uses value mapping to identify the most promising boundaries for exploration to locate instances of a given target object class, and these mappings can be constructed in real time. In contrast, CA-Nav relies on a complete value mapping and subgoal switching mechanism.  Other than these aspects, there do not seem to be significant differences between CA-Nav and VLFM.
2. Details and robustness of instruction decomposition:
You mentioned using a large language model (LLM) to decompose instructions and generate constraints for sub-instructions (e.g. landmarks, locations, directions, etc.). However, the paper does not explain in detail how LLM handles complex or ambiguous instructions. When faced with complex instructions (such as instructions that contain multiple actions or involve multiple constraints simultaneously), how does the system ensure the accuracy of sub-instruction decomposition? Is there a mechanism to handle possible decomposition errors? Can the decomposition process be quantitatively assessed? For example, testing instructions of different complexity for accuracy or robustness.
3. Timing selection for CSM neutron command switching:
The paper mentions that the constraint-aware subcommand manager (CSM) switches subcommands by evaluating the completion of the current constraints. How does CSM determine that a certain sub-instruction has been "completely" completed without switching to the next sub-instruction too early or too late? Does the system have some kind of buffering mechanism to avoid premature switching? Have you considered possible dependencies between subdirectives? Can you share more details about the subcommand switching strategy? For example, how to avoid premature or delayed switching? Have you evaluated the impact of incorrect switching on the overall path planning?
4. The role of CVM’s constraint value map construction and superpixel clustering:
Constraint-aware value mapper (CVM) builds value maps through superpixel clustering to enhance navigation stability. What exactly does superpixel clustering play here? Are there any actual comparative tests to prove that the addition of superpixel clustering brings significant performance improvements? How does CVM perform without superpixel clustering? Can ablation experiments be provided to demonstrate the impact of superpixel clustering on navigation stability and path planning? Furthermore, have other clustering algorithms or image segmentation techniques been considered as an alternative to superpixel clustering?
5. Error recovery mechanism in constraint detection:
The constraint-aware navigation process relies on constraints on landmarks, locations, and directions, and detection errors of these constraints may lead to navigation failure. If a robot detects a landmark incorrectly or fails to recognize constraints correctly, can the system automatically recover? Are there mechanisms in place to handle cases of false detections or incorrect constraint recognition? For example, how to ensure that navigation continues when the robot fails to correctly identify landmarks in the environment? Can you provide more details on the error detection recovery mechanism? For example, when a detection error occurs, does the system re-evaluate the current state or correct it through other visual cues.
6. Generalize to diverse environments and dynamic constraint changes:
Although the method performs well on the R2R-CE and RxR-CE datasets, the environmental diversity of these datasets is relatively limited. Do you plan to test performance in more diverse and dynamic environments? For example, will the challenges that robots need to deal with in dynamic environments (such as landmark occlusion, environmental changes) affect the accuracy of constraint detection? Can you provide more experimental results on dynamic environments? For example, when the environment changes (such as tables and chairs moving or roadblocks appearing), will the agent re-build a value map based on new observations, dynamically adjust constraints or re-evaluate the current navigation goal.

---

> ### Author Response · Authors · 2024-11-21
> **Response to Reviewer xtPJ (Part 1)**
>
> Thank you very much for your review and comments.
>
> ### Q1: Difference from VLFM?
> The difference between CA-Nav and VLFM is two-fold:
> 1. From the task perspective: VLFM addresses Object Navigation, where the agent explores the environment to locate a specific object. CA-Nav, however, focuses on the more complex VLN-CE task, where the agent must follow natural language instructions to navigate. This requires tracking navigation progress and managing sequential sub-tasks.
> 2. From the method perspective: VLFM focuses on frontier-based exploration (FBE), which limits decision options to the boundary of the explored area. In contrast, CA-Nav utilizes the entire value map, offering a larger and more flexible decision space. To efficiently leverage this expanded space, CA-Nav employs a superpixel clustering method for waypoint selection, grouping semantically consistent regions and reducing noise. Additionally, CA-Nav introduces a Constraint-aware Sub-instruction Manager (CSM) to automatically track navigation progress and switch sub-instructions based on predefined constraints, such as landmarks, locations, and directions. To further enhance navigation, CA-Nav applies value map refinements, including trajectory masking and historical decay, ensuring smoother and more robust execution of sequential sub-tasks.
> To the best of our knowledge, this is the first attempt at a constraint-based instruction-switching mechanism in the VLN community.
>
> ### Q2:  How does LLM handle complex or ambiguous instructions? Is there a mechanism to handle possible decomposition errors? Can the decomposition process be quantitatively assessed? For example, testing instructions of different complexity for accuracy or robustness.
>
> 1. As shown in the table below, instructions with more than 5 sub-instructions can be perceived as complex. We show detailed decomposition results of one complex instruction "Go straight. Pass the stairs on the right and continue straight. When you get to the stairs, go up past those as well.  Go into the room with the coaches and then turn right. wait near the glass table with white chairs." The decomposition results are:
>
>   "1313": {"destination": "glass table with white chairs in the room",
>   "sub-instructions": ["Go straight","Pass the stairs on the right and continue straight","When you get to the stairs go up pass those as well","Go into the room with the couches","then turn right","wait near the glass table with white chairs"],
>   "state-constraints": {
>   "0": [["direction constraint","forward"]],
>   "1": [["object constraint","stairs"],["direction constraint","right"]],
>   "2": [["object constraint","stairs"],["direction constraint","up"]],
>   "3": [["location constraint","room"],["object constraint","couches"]],
>   "4": [["direction constraint","right"]],
>   "5": [["object constraint","glass table"],["object constraint","white chairs"]]
>   }}.
>
> It has 6 sub-instructions and some of them have more than one constraint. For example, sub-instruction 1 contains two different constraints and the execution details are depicted in pseudo-code Algorithm1(Line 26) and Algorithm2. Specifically, during sub-instruction 1, at each step, CSM will check both of the two constraints, if one of them has been satisfied it will be removed from the list. CSM switches to the next sub-instruction until all constraints are satisfied.
>
> 2. We didn't design a specific module to handle decomposition errors. However, we established a maximum step threshold to prompt the agent to switch constraints when progress stalls. Similarly, we also set minimum switch steps in order to ensure adequate focus on each constraint before switching.
>
> 3. We add a table to show the quantitative analysis of instruction decomposition. We tested instructions of different complexity for accuracy, and the results are:
> | Number of sub-instructions | 1    | 2    | 3    | 4    | 5    | 6    | 7    |
> |----------------------------|------|------|------|------|------|------|------|
> | success/fail trajectories   | 4/11 | 50/180 | 151/503 | 145/635 | 78/324 | 27/137 | 3/34 |
> | SR                         | 0.36 | 0.28 | 0.30 | 0.23 | 0.24 | 0.20 | 0.09 |
>
> Most instructions are of moderate difficulty with 3 or 4 sub-instructions. In general, the success rate of navigation tends to decrease as the complexity of the instructions increases.

---

> > ### Author Response · Authors · 2024-11-21
> > **Response to Reviewer xtPJ (Part 2)**
> >
> > Thank you very much for your review and comments.
> >
> > ### Q3: Does the system have some kind of buffering mechanism to avoid premature switching? Have you considered possible dependencies between subdirectories? Can you share more details about the subcommand switching strategy? Have you evaluated the impact of incorrect switching on the overall path planning?
> >
> > Yes, we have a buffering mechanism. As shown in Lines 207 - 211, we designed minimum and maximum constraint switch steps to avoid switching too early or too late. We have shown the details in the answer to Q2. Moreover, we evaluated the impact of this mechanism. The results are as follows:
> >
> > | MIN/MAX | 0/0   | 5/25  | 10/25 | 15/25 | 10/15 | 10/35 |
> > |---------|-------|-------|-------|-------|-------|-------|
> > | SR      | 22.3  | 24.1  | 25.3  | 24.6  | 25.0  | 24.4  |
> >
> > The design of the switching buffer mechanism is effective and demonstrates robustness against minor variations in the minimum and maximum constraint switching steps.
> >
> > ### Q4: Are there any actual comparative tests to prove that the addition of superpixel clustering brings significant performance improvements?  How does CVM perform without superpixel clustering? Can ablation experiments be provided to demonstrate the impact of superpixel clustering? Have other clustering algorithms or image segmentation techniques been considered as an alternative to superpixel clustering?
> >
> > We did ablation experiments about the superpixel clustering in Tabel 5. In this table, we compared three different waypoint selection methods: superpixel-based (sr=25.3), frontier-based (sr=21.9), and pixel-based (sr=22.9). Without superpixel, the agent chooses the frontiers or the pixel with the biggest value. However, both frontier-based and pixel-based methods achieve lower sr. After using superpixel-based waypoints, the success rate rises from 21.9 to 25.3, bringing clear improvement.
> > We chose a classical superpixel clustering method, i.e. SLIC, and didn't test other superpixel clustering methods. Alternatively, we analyze the influence of different superpixel sizes:
> >
> > | Superpixel Size | 25*25 | 50*50 | 75*75 | 100*100 |
> > |------------------|-------|-------|-------|---------|
> > | SR               | 23.7  | 24.9  | 24.1  | 21.4    |
> >
> > The results show that CA-Nav's performance begins to degrade when the superpixel size becomes too large. This is because we set the resolution ratio to 5, meaning each pixel corresponds to 5 cm in the real world. When the superpixel size is excessively large, such as 100, it covers a 5m x 5m area, which is too coarse for selecting precise waypoints.
> >
> > ### Q5: Are there mechanisms in place to handle cases of false detections or incorrect constraint recognition?
> > We did not design explicit error recovery mechanisms for constraint detection, primarily because annotating ground truth for each constraint in 3D simulated environments requires significant human effort. However, we have implemented some implicit mechanisms to support error recovery. For instance, our historical decay mechanism calculates a weighted sum of historical and current values. If the robot deviates from the correct path, resulting in the current value being lower than the historical values, it will be guided back to the correct path. In Table 4 we did ablation studies about historical decay mechanism. It improves SR from 22.3 to 24.6.
> >
> > ### Q6: Do you plan to test performance in more diverse and dynamic environments?
> > Navigation in dynamic environments is definitely interesting and important. However, we feel this direction is somehow orthogonal to our work because we mainly focus on sub-instructions switching and value mapping at this stage. To our knowledge, handling dynamic environments may require 3D scene graphs[1], and we plan to incorporate such techniques into our framework in the future.
> >
> > [1] hijie Yan and Shufei Li and Zuoxu Wang and Lixiu Wu and Han Wang and Jun Zhu and Lijiang Chen and Jihong Liu. Dynamic Open-Vocabulary 3D Scene Graphs for Long-term Language-Guided Mobile Manipulation. arXiv preprint arXiv: 2410.11989

---

### Official Review · Reviewer_dEbU · 2024-11-10

**Soundness:** 3
**Presentation:** 3
**Contribution:** 3
**Rating:** 6
**Confidence:** 3

**Summary:**

The paper proposes an LLM based approach for vision and language navigation in continuous environments. The major ideas are two folds: 1. they proposes an LLM-based constraint aware sub-instruction manager which decomposes instructions into location aware direction aware and object aware instructions. Second steps is performed online during navigation i.e. which builds a value map based on the landmark information from first step and segments that into regions. It then selects the most-optimal waypoints by switching to the most relevant sub-instruction.

**Strengths:**

In my opinion, below are the strengths of this paper:

1. Constraints based value map construction idea is promising and offers interpretability.

2. The paper compares to various relevant baselines and shows real world qualitative evaluation results which shows the method can generalize to real-world setting.

4. The paper's presentation is clear, it's nicely written and easy to follow. It also shows a nice gap in zero-shot vision-and-language navigation showcasing it is a promising avenue for future research.

**Weaknesses:**

In my opinion, below are the main weakness of the paper:

1. From Table 1. the results are only slightly better than other zero-shot VLN based approaches, especially for the SPL metric. Can the author discuss this result more?

2. It seems like a heavily engineered approach. Appreciate the authors for including the ablations but it is still not clear if there is one major component that affects the results for the approach. Most of the ablated components (while do impact SR, largely remains the same for SPL and NE).

3. I am wondering if the authors took into account calls to BLIP-2 as well as Grounding-DINO in their inference time calculation i.e. Figure-4? since those would also impact the inference time.

**Questions:**

Please see my questions in the weakness section above.

---

> ### Author Response · Authors · 2024-11-21
> **Response to Reviewer dEbU**
>
> Thank you very much for your review and comments.
>
> ### Q1: From Table 1, why the results are only slightly better than other zero-shot VLN approaches, especially for the SPL metric?
> 1. The baseline A$^2$Nav is not entirely training-free, as it leverages HM3D's room boundaries to train five navigators to handle different actions. In contrast, our approach is completely training-free, making it more challenging. However, compared to the baseline NavGPT-CE (sr=16.3), CA-Nav (sr=25.3) outperformed it by a large margin.
>
> 2. It should be noted that we focus on the zero-shot setting where the agent has not learned prior knowledge of efficient long-horizon planning. We admit that the improvement in SPL is limited because long-horizon planning is a challenging problem, especially when the robot needs to explore unfamiliar environments without prior knowledge. However, this is a common limitation of current zero-shot approaches. We will continue to work on this aspect in the future.
>
> ### Q2: Is there one major component that affects the results of the approach? Why CA-Nav impact SR but largely remain the same for SPL and NE?
> 1. Yes, CSM is the major component, as it not only affects the results but also enables the robot to navigate by following instructions, distinguishing it from simple object navigation. In the VLN community, we are the first to propose a constraint-based instruction-switching mechanism. As shown in Table 3, if we ablate all constraints all metric's performance gets worse.
> 2. Since CA-Nav is a zero-shot method, it can not learn prior knowledge of long-horizon navigation planning like learning-based methods, which often results in longer paths. When the paths are generally longer, SPL becomes less sensitive to changes in path length.
> The SPL metric is calculated as follows:$$\begin{equation}
> SPL = \frac{1}{N}\sum_{i=1}^{N}\mathbb{I}(d(p, p^*)<\epsilon)\frac{L^*}{max(L, L^{*})}
> \end{equation}$$
>
> Where $\mathbb{I}$ is the indicator function, $d(\cdot)$is the distance function, $p$ is the robot's end position, $p^*$ is the ground truth end position, $\epsilon$ is the distance threshold, $L$ is the trajectory length, and $L^*$ is the ground truth trajectory length.
>
> For example, if we test on one success trajectory whose $L^*=12$.
>
> Let agent 1 be a leaning-based method, its trajectory is $L_{1}^{1}=13$ (the subscript represents agent ID, the superscript is experiment ID) and $SPL_{1}^{1}=\frac{12}{13}=0.92$, after ablate some module $L_{1}^{2}=15$, then $SPL_{1}^{2} = \frac{12}{15}=0.8$.
>
> The zero-shot agent 2 may have longer trajectories such as $L_{2}^{1} = 20$, $L_{2}^{2} = 23$, and the results would be: $SPL_{2}^{1} = \frac{12}{20}=0.6$, $SPL_{2}^{2} = \frac{12}{23}=0.52$.
>
> As we can see: $SPL_{1}^{1} - SPL_{1}^{2} = 0.12$; $SPL_{2}^{1} - SPL_{2}^{2} = 0.08$
>
> When the paths are generally longer, SPL becomes less sensitive to changes in path length.
>
>
> ### Q3: If BLIP-2 and Grounding-DINO are taken into account in the inference time calculation?
> Yes, we have already considered BLIP-2 and Grounding-DINO in the time calculation.

---

### Note · Authors · 2024-12-13

I have read and agree with the venue's withdrawal policy on behalf of myself and my co-authors.